# The safety of combined triple drug therapy with ivermectin, diethylcarbamazine and albendazole in the neglected tropical diseases co-endemic setting of Fiji: A cluster randomised trial

Myra Hardy[1,2], Josaia Samuela[3], Mike Kama[3], Meciusela Tuicakau[3], Lucia Romani[4], Margot J. Whitfeld[5], Christopher L. King[6], Gary J. Weil[7], Anneke C. Grobler[1,2], Leanne J. Robinson[8], John M. Kaldor[4], Andrew C. Steer[1,2] *

1 Tropical Diseases Research Group, Murdoch Children's Research Institute, Melbourne, Victoria, Australia, 2 Department of Paediatrics, University of Melbourne, Melbourne, Victoria, Australia, 3 Fiji Ministry of Health and Medical Services, Suva, Fiji, 4 Kirby Institute, University of New South Wales, Sydney, New South Wales, Australia, 5 St Vincent's Hospital, University of New South Wales, Sydney, New South Wales, Australia, 6 Centre for Global Health and Diseases, Case Western Reserve University, Cleveland, Ohio, United States of America, 7 Washington University, St. Louis, Missouri, United States of America, 8 Vector-borne Diseases and Tropical Public Health, Burnet Institute, Melbourne, Victoria, Australia

* andrew.steer@rch.org.au

## Abstract

Lymphatic filariasis has remained endemic in Fiji despite repeated mass drug administration using the well-established and safe combination of diethylcarbamazine and albendazole (DA) since 2002. In certain settings the addition of ivermectin to this combination (IDA) remains a safe strategy and is more efficacious. However, the safety has yet to be described in scabies and soil-transmitted helminth endemic settings like Fiji. Villages of Rotuma and Gau islands were randomised to either DA or IDA. Residents received weight-based treatment unblinded with standard exclusions. Participants were actively found and asked by a nurse about their health daily for the first two days and then asked to seek review for the next five days if unwell. Anyone with severe symptoms were reviewed by a doctor and any serious adverse event was reported to the Medical Monitor and Data Safety Monitoring Board. Of 3612 enrolled and eligible participants, 1216 were randomised to DA and 2396 to IDA. Age and sex in both groups were representative of the population. Over 99% (3598) of participants completed 7 days follow-up. Adverse events were reported by 600 participants (16.7%), distributed equally between treatment groups, with most graded as mild (93.2%). There were three serious adverse events, all judged not attributable to treatment by an independent medical monitor. Fatigue was the most common symptom reported by 8.5%, with headache, dizziness, nausea and arthralgia being the next four most common symptoms. Adverse events were more likely in participants with microfilaremia (43.2% versus 15.7%), but adverse event frequency was not related to the presence of scabies or soil-transmitted helminth infection. IDA has comparable safety to DA with the same frequency of adverse events experienced following community mass drug administration. The presence of co-

**Data Availability Statement:** All relevant data are within the manuscript and its Supporting Information files.

**Funding:** This study was supported in part by grant OPPGH5342 from the Bill & Melinda Gates Foundation to Washington University. The study was also supported in part by the Coalition for Operational Research on Neglected Tropical Diseases, which is funded at the Task Force for Global Health primarily by the Bill & Melinda Gates Foundation, by the United Kingdom Department for International Development, and by the United States Agency for International Development through its Neglected Tropical Diseases Program. Albendazole (produced and donated by GlaxoSmithKline) and diethylcarbamazine (produced and donated by Eisai Co., Ltd.) were obtained from Ministry of Health stocks in Fiji. Ivermectin was purchased from Merck Sharp Dohme (Australia) Pty. Ltd. and permethrin cream 5% was purchased from Pharmatec Wholesale Company Ltd. Fiji and manufactured by Glenmark Pharmaceuticals Ltd. The funders and drug donors had no role in study design, data collection and analysis, decision to publish, or preparation of the manuscript.

**Competing interests:** No authors have competing interests.

endemic infections did not increase adverse events. IDA can be used in community programs where preventative chemotherapy is needed for control of lymphatic filariasis and other neglected tropical diseases.

## Author summary

Lymphatic filariasis is a parasitic infection that is spread between humans by mosquitos. The adult worms can live up to 6 years in humans causing chronic irreversible damage to lymphatic vessels resulting in permanent limb swelling known as elephantiasis. The filariasis worm is susceptible to three different drugs: ivermectin, diethylcarbamazine and albendazole. They have been used in two drug combinations globally in communities at risk of filariasis infection, including Fiji, for over a decade. In an attempt to improve efficacy of the treatment with the ultimate goal of eliminating the infection, the three drugs are now being used in the one administration. In this study, the safety of the triple combination in Fiji was proven to be as safe as the standard two drug treatment. Two other common infections that will be affected by the new treatment, scabies and intestinal worms, did not impact on the frequency of adverse events. The use of the triple combination in Fiji has the potential to improve the control of common neglected diseases without excess side effects.

## Introduction

Lymphatic filariasis is caused by the parasitic, mosquito-borne filarial round-worm *Wuchereria bancrofti* and has been classified by the World Health Organization (WHO) as a neglected tropical disease. The mosquito vector transmits the immature worms, known as microfilariae (Mf), from human to human in 61 endemic countries. The WHO recommends mass drug administration (MDA) for affected communities, as a key element of strategies to control and ultimately eliminate lymphatic filariasis as a public health problem.[1]

Three anti-parasitic drugs, albendazole, diethylcarbamazine and ivermectin, have therapeutic efficacy against lymphatic filariasis and so have been included by WHO in MDA recommendations, with the specific choice of agents dependent on the presence of other endemic pathogens.[2] The precise mechanism of action of these medications on the filarial worm are not fully understood, but efficacy studies have determined macrofilaricidal and microfilaricidal activity when used in either two drug combination. The standard MDA in the Pacific is based on the dual combination of diethylcarbamazine and albendazole (known together as DA). Eleven countries, including Egypt, Thailand and Tonga, have achieved elimination targets and are under surveillance, but others, including eight countries in the Western Pacific Region such as Samoa, Philippines and Fiji, have not done so despite multiple rounds of MDA.[1]

Elephantiasis, one of the severe complications of lymphatic filariasis, was first reported from Fiji in 1841, on the island of Rotuma [3], and a national survey in the 1940s observed a Mf prevalence of 12.7%.[4] After limited success with vector control [5, 6], the Ministry of Health commenced a national MDA with diethylcarbamazine in 1969.[7] In 1999, Fiji joined the Pacific Programme to Eliminate Lymphatic Filariasis and delivered annual MDA from 2002 using DA. However, despite reportedly adequate DA treatment coverage above 65%, lymphatic filariasis has remained endemic in specific areas of Fiji.[1]

Combining the three active agents into the one MDA combination, known as IDA, does not create any drug-drug interactions, and is superior to DA in clearing microfilaremia at least as long as 36 months after a single round of treatment.[8–10] With more effective filariasis treatments it is expected that persons with microfilaremia will experience more adverse events (AEs), attributable to the response to dead or dying parasites.[11, 12] This effect on AEs was reported from a small efficacy trial based in Papua New Guinea.[8, 9] Fiji joined a 5-site international collaboration to examine the effect of IDA in a larger number of participants across multiple geographic settings. This large global study of IDA confirmed more AEs occurred, but the majority were mild and transient, and that this combination was a safe strategy for community control of filariasis.[13, 14]

Ivermectin is also efficacious against two other neglected tropical diseases endemic in Fiji, scabies and soil-transmitted helminths (STH).[2, 15–18] A national survey of scabies in 2007 found that all age groups and regions of Fiji are affected, with an estimated national prevalence of 18.5%.[16, 19] Soil-transmitted helminths are also common in Fiji with a prevalence as high as 45% in school-aged children.[20–22] Because the medications in the IDA drug combination have activity against scabies and STH in addition to lymphatic filariasis, MDA with IDA represents an integrated intervention against three neglected tropical diseases. Successful treatment of scabies and STH is expected to induce mild and transient symptoms primarily pruritus (scabies) and abdominal pain (STH).[16, 18] We aimed to evaluate the safety of IDA in Fiji, providing a detailed account of AEs following DA and IDA mass drug administration in Fiji and an analysis of factors associated with their occurrence including filariasis, scabies, and STH infections.

## Methods

We conducted a cluster randomised trial to compare the safety of MDA with DA and IDA. We chose a cluster randomised design because the intervention is implemented at the community level. The study procedures as applied to individuals were consistent with the protocol of the multi-site safety trial of IDA (S1 Protocol).[13]

### Ethical approvals, trial registration and oversight

Relevant government ministries and district offices in Fiji approved the engagement of communities on these islands. The protocol was reviewed by the Royal Children's Hospital Melbourne Human Research Ethics Committee (reference 36205) and Fiji National Health Research and Ethics Review Committee (reference 2016.81.MC). The trial was registered (Clinitrials.gov NCT03177993 and ANZCTR N12617000738325). An independent Data Safety Monitoring Board of six experts reviewed the protocol and met periodically throughout the study to review reports.

### Setting

The study was conducted in Fiji, an island nation in the Southwest Pacific with a population in the most recent census (2017) of 884,887, a median age of 27.5 years and with 56% located in urban settings.[23] People are predominantly of Melanesian, Polynesian or Indian descent. There are four governmental divisions: Central, Northern, Eastern, and Western.

The Eastern Division was chosen for the study because it has the highest Mf prevalence in the country of 2.2% in 2007. [24] Two of the divisions islands from this region, Rotuma and Gau, were selected study sites. Both had previously received at least 10 rounds of MDA and, based on filarial antigen prevalence, were due for a further round in 2017.[25] The islands met

other criteria for being study sites, including having reliable communications, a hospital, and an airstrip for medical evacuation.

## Trial design

Villages were designated as clusters. All 35 villages on the two islands agreed to participate in this study prior to randomisation. An independent statistician generated randomised treatment allocation using Stata software in a 1:1:1 ratio stratified by island to either DA, IDA1 (ivermectin administered with diethylcarbamazine and albendazole) or IDA2 (same as IDA1, plus a second dose of ivermectin on day 8). The IDA2 group was included to evaluate the community effectiveness of one versus two doses of ivermectin not reported here. Our study design differed from the global study by including all community members regardless of eligibility for LF treatment in order to allow community assessments for all three infections at both timepoints. For the purpose of analysing the safety of IDA up to 7 days, the IDA1 and IDA2 groups were identical, so they were combined for safety analyses.

## Treatments

Albendazole was provided as a fixed oral dose of 400mg. Diethylcarbamazine and ivermectin were dosed according to weight and whole tablet ranges, aiming for 6mg/kg and 200μg/kg respectively, (S1 Table). Participants and assessors were unblinded to treatment given.

Participants were excluded from receiving either IDA or DA treatment if they were less than 2 years of age or 15 kg in weight, were pregnant, breastfeeding within 7 days of delivery, or known to have a severe illness or allergy to study drugs. All women of child-bearing age were asked on the timing of their last menses. If menses was more than 4 weeks or unknown, a urine pregnancy test was offered. No contraceptive measures were provided or recommended, consistent with WHO policy for LF MDA. Additionally, those in villages allocated to IDA were excluded from taking ivermectin if aged less than 5 years. Topical permethrin cream 5%, which is active against scabies, was provided to all individuals excluded from taking ivermectin. It was also offered to individuals in the DA group identified at baseline screening as having scabies and to their household contacts (S1 Fig).

All oral medication was taken under direct observation by a study team member. Permethrin application was not observed. Instructions to adults in affected households recommended permethrin to be applied over the whole body for 8 hours overnight (4 hours for children less than 2 months of age) before washing off.

## Community engagement and enrolment

Prior to enrolment there was extensive community engagement in each village including liaison with leaders and health representatives, and an interactive presentation. This introduced and explained the study, the treatment group to which their village had been allocated, expected date of enrolment, and the different procedures on enrolment day: individual consent, screening for infections and administration of study medications. Anyone living in the village at the time of MDA was invited to the community central meeting place to enrol regardless of eligibility for MDA. Outreach visits to homes were conducted to allow people with poor mobility to participate. Participants 13 years and older were asked to provide written consent. Parental/guardian consent was required for all people aged less than 18 years and for adults without capacity to consent. For individuals 13–17 years dual consent was required. Written material was provided in English and the Fijian language, iTaukei, and staff members provided verbal translation in iTaukei and Rotuman as required. Representatives from each

village assisted the study team in creating up-to-date lists of village residents by household and reasons for non-participation: temporarily away or declined.

## Safety assessments

The primary outcome of safety of IDA was evaluated at the individual and cluster levels. A training package from the global study [13] was provided to all staff to ensure consistency of reporting between staff and across sites. Safety was assessed in two periods, consistent with the timing of AEs observed in previous studies.[8–11] Active follow-up of all participants occurred daily in the first two days following treatment, when more severe symptoms are expected with death of microfilariae.[11] For continuity, participants were seen by the same nurse on the day of taking tablets and day 1 and day 2 afterwards, and asked a standard open-ended question about their health. During days 3 to 7 after treatment, monitoring comprised of two activities: 1) participants were asked to notify a study representative if they were unwell and required assessment, and 2) participants previously identified as having an AE judged moderate or worse were assessed each day, with assessments only stopping if symptoms improved to mild severity or resolved completely (S2 Fig).

Symptoms were classified using terms from Medical Dictionary for Regulatory Activities [26], and were graded based on the impact on daily activities of the participant: from mild (no impact); moderate (unable to engage in normal functional activities like work or school); severe (unable to undertake personal activities of daily living); or life-threatening (S2 Table). [27] Further, participants identified with an AE judged moderate or worse had vital signs taken by the study nurse. Participants with events graded as severe or life-threatening, or graded as moderate with abnormal vital signs, symptoms of unclear cause or generating nurse concern, were reviewed by a local doctor for full examination, management and assessment of whether the event met the criteria for a serious AE (S2 Fig).

A serious AE was defined as one resulting in at least one of the following outcomes: death, life-threatening event, new or prolongation of a hospital admission, persistent or significant disability/incapacity, congenital anomaly, or other serious health event requiring medical intervention to prevent one of the other outcomes listed above. All serious AEs were discussed with the Medical Monitor, an offsite independent senior doctor trained in safety reporting. The Global Medical Monitor reviewed all serious AE reports before they were reported to the Data Safety Monitoring Board, ethics committees and medication suppliers.

## Parasitology assessments

Participants were tested at baseline for the presence of parasitic infections known to be sensitive to one or more of the study drugs. From an efficacy perspective, this testing would serve as a baseline for subsequent assessments. From a safety perspective, it was important to be able to relate potential effects of co-infection on drug efficacy and AEs.

**Lymphatic filariasis.** The prevalence of filarial infection was determined by two standard methods. Circulating filarial antigen (CFA) was detected by placing approximately 75 μl of capillary blood onto the rapid diagnostic test, Alere Filariasis Test Strip (FTS, Alere Scarborough, Inc., Scarborough, ME, USA). The CFA was quantified by comparing the colour strength of the test strip against the control line at 10 minutes, scored as: CFA1 (weak positive); CFA2 (medium positive); and CFA3 (strong positive).[28] The second method detects active infection with *W. bancrofti* by light microscopy for Mf in a 60 μl stained capillary blood smear. Mf testing was only performed for participants with a positive CFA result only. Each blood smear was read independently by two laboratory technicians.[29, 30]

**Scabies.**   Scabies was diagnosed by trained nurses by identifying pruritic inflammatory papules in a typical distribution.[31]

**Soil-transmitted helminths.**   Stool pots and collection kits were distributed to the community prior to enrolment, and participants asked to bring in fresh stool on the day of enrolment, and prior to administration of study drug. Stool was processed by the Kato-Katz method within 12 hours of collection.[32] Smears were prepared in duplicate with each one read independently by one of two trained laboratory technicians. A stool sample was considered positive if either technician identified eggs of *Ascaris*, *Trichuris* or hookworm.

## Data management

Data were captured in real time into an electronic data capture system developed specifically for the Fiji trial site by CliniOps (Fremont, CA, USA). The system allowed study team members to enter de-identified information at enrolment for each participant, and linked information collected at subsequent visits to the participant's record.

## Statistical methods

We reported our results consistent with Consolidated Standards of Reporting Trials (CONSORT) recommendations for cluster trials (S3 Table).[33, 34] Data were analysed as prospectively planned and in accordance with International Conference on Harmonisation Statistical Principles for Clinical Trials Guideline E9. A statistical analysis plan was written for the main study and followed for the Fiji study. The Fiji site differs from the global study analysis by including children aged 2–4 years who received LF treatment with DA according to WHO guidelines. We described AEs by absolute number, severity, symptoms, treatment group, island, sex, age-group and parasitological status at baseline. We used logistic regression adjusted for clustering by village and stratification by island for statistical significance. We compared AEs between subgroups in univariate and multivariate generalised linear models, adjusting for clustering by village and stratification by island. We did not include STH results in the multivariate analysis due to the low number of processed samples. Filariasis infection was represented only by Mf results for the multivariate analysis. For sub-group analyses with small denominators, statistical significance has been determined using Fischer's exact test without any adjustment for village clustering or island stratification.

**Sample size.**   Fiji was part of a multi-site study (other sites included Haiti, India, Indonesia and Papua New Guinea) that enrolled 26,836 participants 5 years and older.[13] This sample size is powered to detect serious AEs with a frequency of less than 0.1%. This level of serious AE detection is recommended by WHO for endorsement of new public health treatments.[35] The individual sites are not powered for this primary outcome. The sample size of groups in Fiji was adequate to evaluate the effect of one versus two doses of ivermectin on scabies, not reported here.

## Results

### Enrolment

The total resident population of the two islands was 4610: 1994 in 17 villages on Rotuma and 2616 in 18 villages on Gau (Fig 1, Table 1). The median village size was 108 people, range 18 to 298 (S4 Table). Participants were recruited over a four-month period from July 2017 to November 2017. The total number consented was 3812 (82.7% enrolment coverage). There was 80% participation for the villages randomised for DA treatment and 84.1% for those randomized for IDA (85.9% for IDA1 and 82.6% for IDA2). There were 449 people (9.7%)

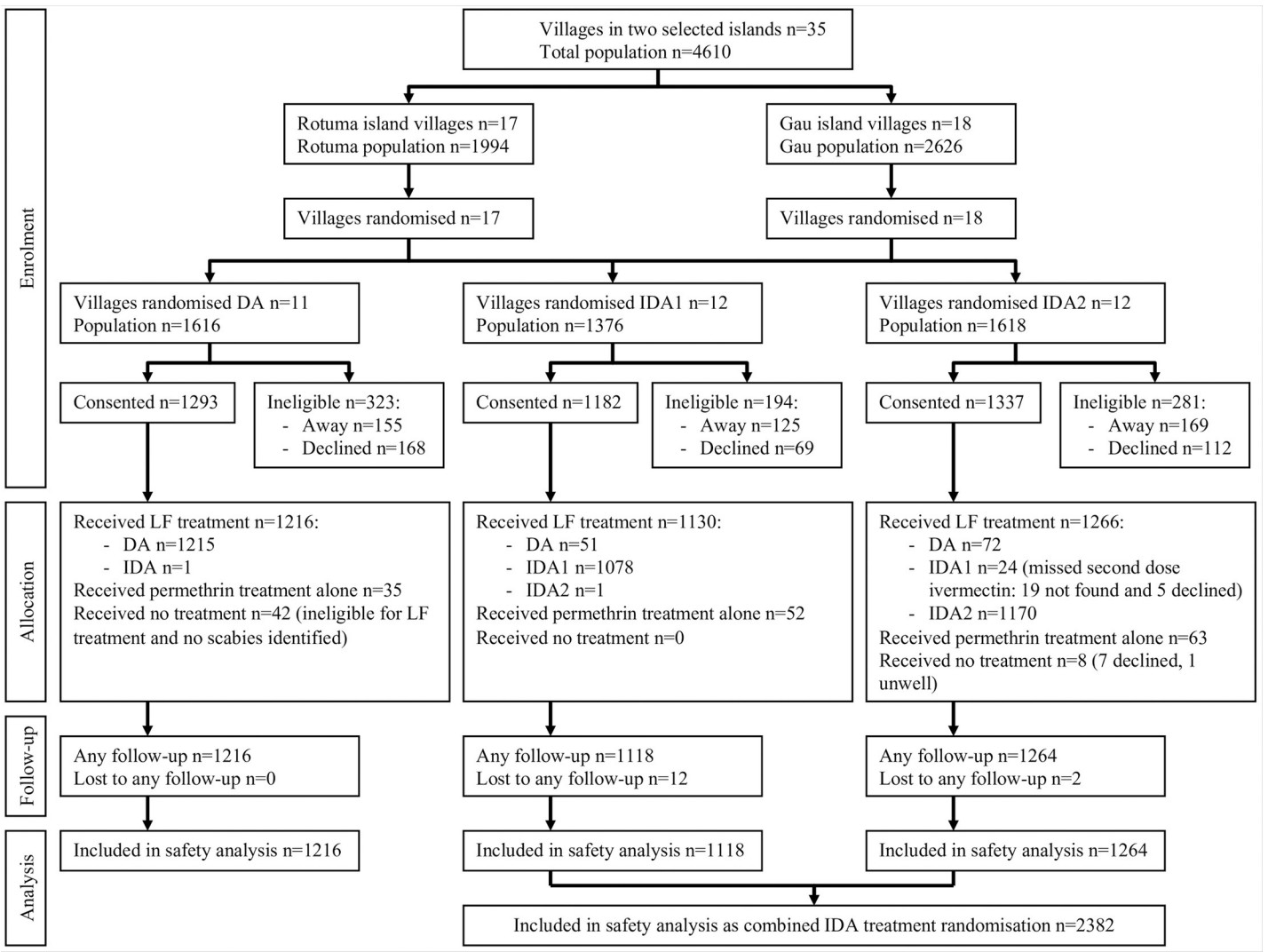

**Fig 1. CONSORT flow diagram detailing village cluster randomisation, individual enrolment, treatment received, safety monitoring follow-up and resulting total including in analyses.** DA: diethylcarbamazine and albendazole; IDA1: ivermectin one dose, diethylcarbamazine and albendazole; IDA2: ivermectin two doses, diethylcarbamazine and albendazole.

temporarily away from their village at the time of the enrolment visit and 349 people (7.6%) who declined (DA group 10.4%, IDA group 6%, S5 Table). The age and sex distribution of enrolled participants was similar to the overall population distribution of the two islands (S4 Table). The median height and weight for the 2161 participants aged 18 years and older was 169 cm (IQR 163–175 cm) and 84.7 kg (IQR 72.5–97.7 kg) respectively.

## Parasitic infections

Lymphatic filariasis assessments were conducted on 3659 of the eligible 3719 participants aged 2 years and above (98.4%). The reasons for non-testing were: declined blood testing (n = 60); declined a repeat bleed for smear preparation (n = 11); or smear was unreadable due to technical difficulties (n = 11). The prevalence of lymphatic filariasis across the two treatment groups was comparable as measured by detection of CFA (14.1%, 95% CI 11.3–17.5%) and Mf (3.8%, 95% CI 2.6–5.6%, Table 1, S6 Table). The geometric mean Mf density was also similar between

**Table 1. Participant demographic characteristics and baseline parasitic infection prevalence by treatment group.**

| | | DA | | | IDA | | | Total | | |
|---|---|---|---|---|---|---|---|---|---|---|
| | | n | % | (95% CI) | n | % | (95% CI) | n | % | (95% CI) |
| Enrolment | Total population | 1616 | | | 2994 | | | 4610 | | |
| | Total consented | 1293 | 80.0 | | 2519 | 84.1 | | 3812 | 82.7 | |
| Sex | Male | 673 | 52.0 | (49.7–54.4) | 1293 | 51.3 | (49.4–53.2) | 1966 | 51.6 | (50.1–53.1) |
| | Female | 620 | 48.0 | (45.6–50.3) | 1226 | 48.7 | (46.8–50.6) | 1846 | 48.4 | (46.9–49.9) |
| Age (years) | Median (IQR) | 27 | | (10–48) | 24 | | (10–46) | 25 | | (10–47) |
| | <2 | 38 | 2.9 | (2.1–4.0) | 55 | 2.2 | (1.5–3.1) | 93 | 2.4 | (1.9–3.1) |
| | 2–4 | 80 | 6.2 | (5.3–7.3) | 168 | 6.7 | (5.6–7.9) | 248 | 6.5 | (5.7–7.4) |
| | 5–9 | 182 | 14.1 | (11.4–17.3) | 359 | 14.3 | (12.3–16.5) | 541 | 14.2 | (12.6–16.0) |
| | 10–14 | 172 | 13.3 | (10.6–16.6) | 359 | 14.3 | (12.2–16.6) | 531 | 13.9 | (12.3–15.8) |
| | 15–24 | 134 | 10.4 | (8.2–13.0) | 328 | 13.0 | (7.6–21.4) | 462 | 12.1 | (8.2–17.5) |
| | 25–34 | 154 | 11.9 | (9.3–15.1) | 264 | 10.5 | (9.2–11.9) | 418 | 11.0 | (9.7–12.3) |
| | 35–49 | 236 | 18.3 | (15.8–21.0) | 449 | 17.8 | (15.5–20.4) | 685 | 18.0 | (16.2–19.9) |
| | 50–64 | 207 | 16.0 | (12.7–20.0) | 358 | 14.2 | (12.0–16.8) | 565 | 14.8 | (13.0–16.9) |
| | ≥65 | 90 | 7.0 | (5.8–8.3) | 179 | 7.1 | (5.6–8.9) | 269 | 7.1 | (6.0–8.3) |
| LF | Total assessed CFA | 1239 | | | 2420 | | | 3659 | | |
| | Total CFA positive | 186 | 15.0 | (8.8–24.4) | 330 | 13.6 | (10.0–18.3) | 516 | 14.1 | (11.3–17.5) |
| | CFA1—weak positive | 64 | 5.2 | (3.7–7.2) | 107 | 4.4 | (3.2–6.1) | 171 | 4.7 | (3.8–5.8) |
| | CFA2—medium positive | 59 | 4.8 | (2.4–9.4) | 98 | 4.1 | (3.0–5.4) | 157 | 4.3 | (3.3–5.5) |
| | CFA3—strong positive | 63 | 5.1 | (2.1–11.6) | 125 | 5.2 | (3.4–7.8) | 188 | 5.1 | (3.6–7.4) |
| LF | Microfilariae positive[a] | 47 | 3.8 | (1.5–9.1) | 93 | 3.9 | (2.6–5.8) | 140 | 3.8 | (2.6–5.6) |
| | Geometric mean density Mf/ml | 239 | | (149–383) | 183 | | (128–260) | 200 | | (151–264) |
| Scabies | Identified[b] | 176 | 13.6 | (7.9–22.4) | 337 | 13.4 | (9.9–17.9) | 513 | 13.5 | (10.4–17.3) |
| STH | Stool specimen assessed | 268 | 20.7 | | 658 | 26.1 | | 926 | 24.3 | |
| | Positive microscopy | 35 | 13.1 | (6.3–25.1) | 136 | 20.7 | (13.2–30.9) | 171 | 18.5 | (12.7–26.1) |
| LF Treatment[c] | | 1216 | 94.0 | (92.5–95.3) | 2396 | 95.1 | (93.9–96.1) | 3612 | 94.8 | (93.9–95.5) |
| Any follow-up after LF treatment | | 1216 | 100 | - | 2382 | 99.4 | (98.5–99.8) | 3598 | 99.6 | (99.0–99.9) |

DA: diethylcarbamazine and albendazole; IDA: ivermectin, diethylcarbamazine and albendazole; IQR: interquartile range; LF: lymphatic filariasis; CFA: circulating filarial antigen; STH: soil-transmitted helminths; CI: confidence intervals

[a] Denominator is equal to total assessed CFA subtracting 22 with smears declined or unreadable (DA N = 1238, IDA N = 2399, Total N = 3637).

[b] Denominator is equal to total consented for each group.

[c] Denominator is equal to total consented for each group. Participants received treatment as randomised except 1 in DA received IDA, 123 in IDA received DA.

groups (DA 239 Mf/ml, range 17–4643, 95% CI 149–383, versus IDA 183 Mf/ml, range 17–9168, 95% CI 128–260). The intracluster correlation coefficient for Mf was 0.205. Rotuma had a higher CFA prevalence (24.8% versus Gau 6.1%; risk difference (RD) 19.3%, 95% CI 14.1–24.5%) and Mf prevalence (6.9% versus 1.6%; RD 5.6%, 95% CI 2.9–8.2%), but a comparable infection intensity (geometric mean Mf density Rotuma 206 Mf/ml, range 17–9168, 95% CI 148–288, versus Gau 181 Mf/ml, range 17–4492, 95% CI 106–308). Mf prevalence was higher in males (6% versus females 1.6%; RD 4.4%, 95% CI 2.4–6.3%), and those aged 35–49 years had the highest Mf prevalence (7.8%, 95% CI 4.9–12.3%). As the CFA score increased, the proportion with a positive test for Mf also significantly increased; 4.8% of grade 1 CFA were positive for Mf compared to 55.6% of those with CFA scores of 3 (RD 50.6%, 95% CI 41.5–59.6%).

Scabies was evenly distributed across treatment groups with an overall prevalence of 13.5% (95% CI 10.4–17.3%), and prevalence peaking at 32.9% (95% CI 25.7–41.0%) in the children aged 5–9 years. It was more common on Gau (16.9%, 95% CI 12.6–22.2%, versus Rotuma

8.8%, 95% CI 5.6–13.7%, RD 8.5%, 95% CI 2.7–14.4%) and in females (15% versus males 12%, RD 3.0%, 95% CI 1.1–4.9%). The STH prevalence was higher in the IDA groups (20.7%, 95% CI 13.2–30.9%) compared to the DA group (13.1%, 95% CI 6.3–25.1%, RD 5.3%, 95% CI 0–10.5%). Gau had a higher STH prevalence (25.7%, 95% CI 16.9–37.0%, versus Rotuma 5.0%, 95% CI 2.8–8.8%, RD 20.1%, 95% CI 10.8–29.5%), as did males (23.5%, 95% CI 17.1–31.6%, versus females 13.8%, 95% CI 8.2–22.2%, RD 10.9%, 95%CI 5.3–16.4%). Children aged 10–14 years had the highest STH prevalence (30.1%, 95% CI 13.3–54.6%).

One infection (either filariasis, scabies or STH), was identified in 1136 participants (29.8% of enrolled). Sixty-one participants had two infections identified and three participants were co-infected with all three.

## Treatment

A majority of participants were eligible and received filariasis treatment according to the assignment of their village (94% and 95.1% in DA and IDA groups respectively). One person in the DA group received IDA and 123 in the IDA group received DA (due to exclusion from ivermectin because of weight and/or age). Thirty-two of the 305 (10.5%) participants with scabies in the IDA group were ineligible for ivermectin and received permethrin instead on day 0. The 164 participants with scabies in the DA group did not receive permethrin treatment until day 8, after safety monitoring period was completed. Applying intention to treat principles, we included these individuals for analysis in their village randomisation groups and not by the individual treatment they received.

## Adverse events

Of the 3612 participants who received MDA, 14 (0.4%) were lost to follow-up during the 7-day safety monitoring period following treatment (Fig 1). One participant was not seen during the two-day active monitoring period but presented to the study nurse for review on day 5.

One or more AE was reported by 600 (16.7%) participants (Table 2, S6 Table), with symptoms first starting during the active monitoring period for 92.7%. AE rates did not differ by treatment group, and similar AE frequencies were reported after the two treatments in males (DA 15.9% versus IDA 15.1%, $P = 0.80$) and in persons with microfilaremia (DA 45.7% versus IDA 41.9%, $P = 0.34$). Participants found to have scabies on examination reported AEs less frequently than those without (14.3% vs 17.0%, $P = 0.24$). This difference was most marked in the DA group (12.2% AEs in participants with scabies versus 17.4% without scabies, $P = 0.04$) compared to the IDA group (15.4% with scabies versus 16.9% without, $P = 0.67$). There were 121 children aged less than 5 years in the IDA group that received DA and permethrin and were followed up. Of these, six (5.0%) experienced an AE. There were 32 that had scabies and two (6.3%) experienced an AE. In contrast, there were 56 children aged less than 5 years in the DA group that received DA and were followed up. Six (10.7%) experienced an AE. None of the 12 participants with scabies in this sub-group reported an AE, noting that they also did not receive scabies directed treatment until day 8. We found a similar difference in AE reporting based on STH results (13.9% AEs with STH positive stool versus 18.4% STH negative, $P = 0.13$). We observed a similar frequency of AEs when we excluded children aged less than 5 years (DA 17.0% versus IDA 17.3%, $P = 0.90$).

The type and severity of AEs was similar across treatment groups (Fig 2, S7 Table). Fatigue was the most common AE with 307 (51.2%) of participants who experienced an AE reporting this symptom. Of reported events, 93.2% were graded as mild, 5.5% moderate and 1.3% severe. Symptoms reported as severe by 8 participants (0.2% of participants treated and followed up) were fatigue, headache, dizziness, myalgia, diarrhoea, rash, dyspnoea and chills. There was no

**Table 2. Frequency of adverse events by treatment group.**

| | | DA | | | IDA | | | Total | | |
|---|---|---|---|---|---|---|---|---|---|---|
| | | N | n | % | N | n | % | N | n | % |
| Any AE | Total treated LF and followed up[b] | 1216 | 203 | 16.7 | 2382 | 397 | 16.7 | 3598 | 600 | 16.7 |
| Island | Rotuma | 412 | 55 | 13.3 | 1152 | 203 | 17.6 | 1564 | 258 | 16.5 |
| | Gau | 804 | 148 | 18.4 | 1230 | 194 | 15.8 | 2034 | 342 | 16.8 |
| Sex | Male | 642 | 102 | 15.9 | 1241 | 188 | 15.1 | 1883 | 290 | 15.4 |
| | Female | 574 | 101 | 17.6 | 1141 | 209 | 18.3 | 1715 | 310 | 18.1 |
| Age (years) | Reported AE median (IQR) | 41 | (19.5–55) | | 33 | (15–49) | | 36 | (16–51.5) | |
| | 2–4 | 56 | 6 | 10.7 | 123 | 6 | 4.9 | 179 | 12 | 6.7 |
| | 5–9 | 181 | 22 | 12.2 | 356 | 36 | 10.1 | 537 | 58 | 10.8 |
| | 10–14 | 172 | 13 | 7.6 | 358 | 48 | 13.4 | 530 | 61 | 11.5 |
| | 15–24 | 132 | 17 | 12.9 | 322 | 60 | 18.6 | 454 | 77 | 17.0 |
| | 25–34 | 146 | 27 | 18.5 | 247 | 53 | 21.5 | 393 | 80 | 20.4 |
| | 35–49 | 235 | 50 | 21.3 | 442 | 96 | 21.7 | 677 | 146 | 21.6 |
| | 50–64 | 206 | 50 | 24.3 | 358 | 69 | 19.3 | 564 | 119 | 21.1 |
| | ≥65 | 88 | 18 | 20.5 | 176 | 29 | 16.5 | 264 | 47 | 17.8 |
| LF[a] | CFA negative | 1030 | 155 | 15.0 | 2035 | 307 | 15.1 | 3065 | 462 | 15.1 |
| | Total CFA positive | 184 | 48 | 26.1 | 328 | 89 | 27.1 | 512 | 137 | 26.8 |
| | CFA1—weak positive | 63 | 14 | 22.2 | 107 | 22 | 20.6 | 170 | 36 | 21.2 |
| | CFA2—medium positive | 59 | 15 | 25.4 | 97 | 25 | 25.8 | 156 | 40 | 25.6 |
| | CFA3—strong positive | 62 | 19 | 30.6 | 124 | 42 | 33.9 | 186 | 61 | 32.8 |
| LF | Microfilariae negative | 1167 | 182 | 15.6 | 2249 | 354 | 15.7 | 3416 | 536 | 15.7 |
| | Microfilariae positive | 46 | 21 | 45.7 | 93 | 39 | 41.9 | 139 | 60 | 43.2 |
| Scabies | Not identified | 1052 | 183 | 17.4 | 2077 | 350 | 16.9 | 3129 | 533 | 17.0 |
| | Identified | 164 | 20 | 12.2 | 305 | 47 | 15.4 | 469 | 67 | 14.3 |
| STH | Negative microscopy | 226 | 40 | 17.7 | 504 | 94 | 18.7 | 730 | 134 | 18.4 |
| | Positive microscopy | 35 | 5 | 14.3 | 131 | 18 | 13.7 | 166 | 23 | 13.9 |

DA: diethylcarbamazine and albendazole; IDA: ivermectin, diethylcarbamazine and albendazole; AE: adverse event; IQR: interquartile range; LF: lymphatic filariasis; CFA: circulating filarial antigen; STH: soil-transmitted helminths.

[a] Total assessed for LF CFA N = 3577 (2 in DA and 19 in IDA weren't assessed for LF).

[b] Participants received treatment as randomised except 1 in DA village received IDA, and 121 in IDA villages received DA (ineligible for ivermectin due to weight and/or age).

life-threatening AE in either treatment group. Eighteen (3.2%) had an AE that persisted beyond 48 hrs following treatment and 44 (7.3%) experienced their first AE after 48 hrs. Only two participants had persistent AEs with severity greater than mild by day 7 after treatment.

In the 139 participants with microfilaremia that were treated, arthralgia (n = 8, 17.4%) and fatigue (n = 7, 15.2%) were the most common symptoms experienced in the DA group. Whereas fatigue (n = 23, 24.7%), arthralgia (n = 13, 14.0%), and muscle weakness (n = 12, 12.9%) were the most common in the IDA group. Fever and scrotal pain, two other commonly reported symptoms following filariasis treatment in the presence of microfilaremia, were reported in 2.9% (n = 3, 6.5% in DA and n = 1, 1.1% in IDA group, $P = 0.11$) and 2.2% (equal in both groups) respectively. There was only one participant in each treatment group with a severe AE. In a post-hoc analysis of participants with microfilaremia, the geometric mean Mf density was significantly higher in those that experienced an AE after treatment (357 Mf/ml, 95% CI 223–569 versus 131 Mf/ml, 95% CI 95–181, ttest $P = 0.0004$).

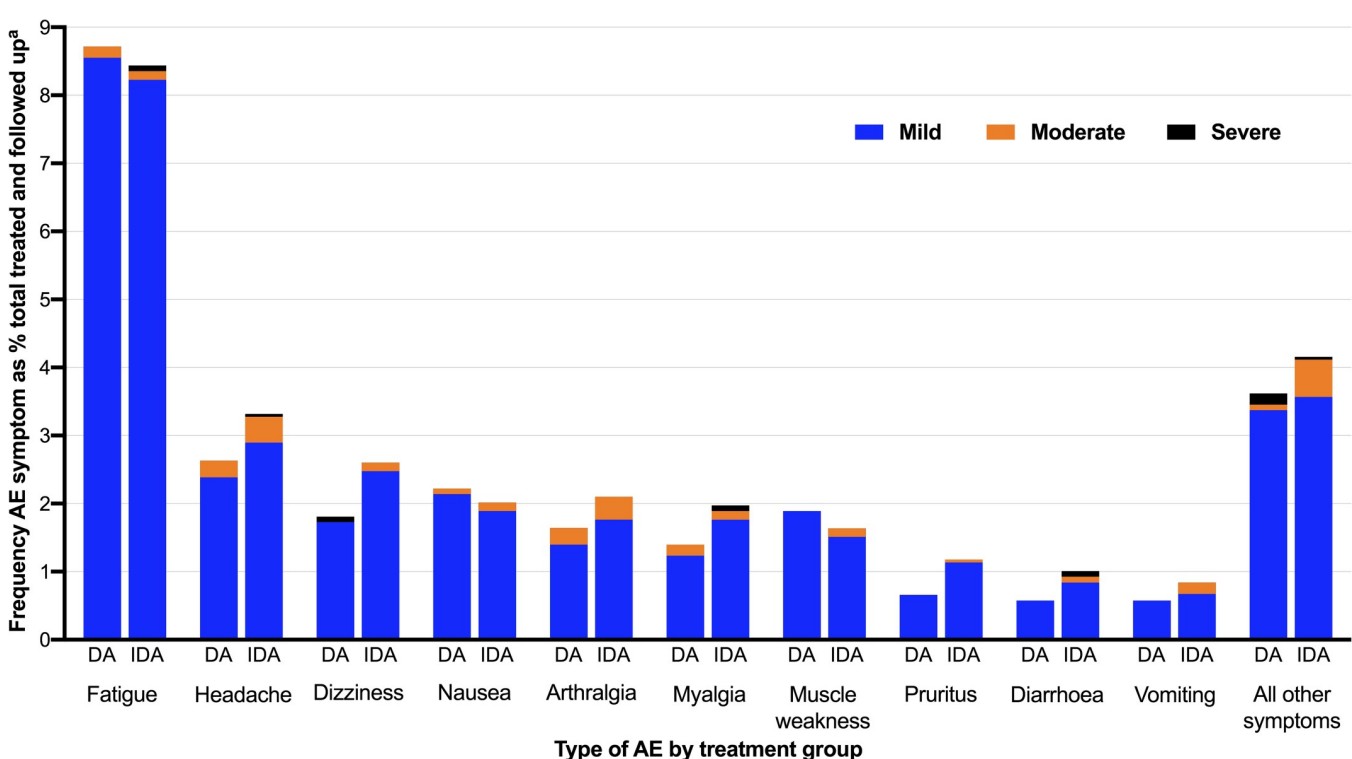

**Fig 2. Type, frequency and severity of adverse event symptoms by treatment group, reported as a percentage of total number of participants treated and followed up.** DA: diethylcarbamazine and albendazole; IDA: ivermectin, diethylcarbamazine and albendazole; AE: adverse event. [a] Denominator total treated and followed up N = 3598.

In the 469 participants with scabies that were treated for filariasis, 67 (14.3%) reported an AE, and a higher proportion were in the IDA group (n = 47, 15.4% versus DA n = 20, 12.2%, $P = 0.61$). In this scabies sub-group, fatigue remained the most common symptom (DA n = 10, 6.1% versus IDA n = 23, 7.5%, $P = 0.71$). Abdominal pain was reported more in both treatment groups with scabies (DA n = 2, 1.2% and IDA n = 3, 1.0%) compared to all participants (n = 18, 0.5%, $P = 0.22$). In the IDA group with scabies compared to the whole cohort, there was notable increased reporting of pruritus (n = 7, 2.3% versus n = 36, 1.0%, $P = 0.046$ respectively) and rash (n = 5, 1.6% versus n = 15, 0.4%, $P = 0.016$ respectively). Only one participant who reported pruritus in the IDA group received permethrin.

An STH infection was identified in 166 participants and AEs were reported by 23 (13.9%). Gastrointestinal symptoms (nausea, vomiting, diarrhoea, abdominal pain, and abdominal distension) were reported less by members of this sub-group than overall (STH positive DA 0% versus IDA n = 5, 3.8%, versus overall n = 153, 4.3%).

## Serious adverse events

There were three serious AE reports. One participant who received DA, had dizziness and hypertension requiring overnight hospitalization, six days after treatment, and another in the DA group had an acute exacerbation of chronic obstructive pulmonary disease requiring hospitalization, two days after treatment. One participant with pre-existing lower limb lymphoedema developed cellulitis of the limb five days after IDA treatment. This participant experienced severe limitation of mobility and personal activities of daily living, but did not require hospitalization. All three had complete resolution of their symptoms.

### Risk factors for adverse events

Univariate and multivariate analyses revealed no difference in reporting of AEs between treatment groups or by island locations (Table 3). Participants aged 50–64 years of age had the highest risk of an AE (adjusted RD 12.5%, 95% CI 5.4–19.6%). We observed a linear relationship between baseline filarial infection status of participants and AE reporting, with 15.1% of participants having a negative CFA test reporting an AE compared to 33.1% of those with a grade 3 CFA test (RD 19.4%, 95% CI 10.5–28.4%). Further, 43.2% of participants with Mf reported an AE compared to 15.8% of those without Mf (adjusted RD 26.4%, 95% CI 18.5–34.3%). Males reported fewer AEs (15.4%) than females (18.1%, adjusted RD 3.5%, 95% CI 0.9–6.1%), despite having a much higher Mf prevalence (6% versus 1.6%). There was no increased risk of reporting of AEs in participants with scabies or STH (Tables 2 and 3).

## Discussion

Our study demonstrates that community MDA with IDA has a similar AE profile to the traditional DA combination in Fiji. Approximately one in every six individuals treated experienced an AE, but the vast majority of these events were mild and resolved within seven days. Fatigue, headache, dizziness, nausea and arthralgia were the five most common symptoms reported. The type of AEs in our study were consistent with those reported in previous studies.[8, 13, 36]

The safety of IDA has been demonstrated in small randomised trials[8, 10], and now in a large multi-site community safety trial that included Fiji.[13] However, these studies have not reported on the effect of scabies and STH infections on the reporting of AEs. The Fiji results we report here, clearly outline the community prevalence of each neglected tropical disease and their relationship to reporting of AEs following IDA compared to DA. Despite a high burden of scabies and STH there was no increased reporting of AEs in persons with these infections.

As expected, we observed a strong positive relationship between markers of filarial infection and the frequency of AEs.[11] This was most prominent among participants with microfilaremia detected by the 60 μl smear. However, it was also observed in participants who were CFA positive without microfilaremia. Since most AEs are attributed to death of Mf [11], it is possible that these participants had low density microfilaremia that was not detected by smear, or Mf were not circulating at the time of testing. This uncertainty around presence of Mf might have been avoided if we used the more sensitive method of membrane filtration with 1ml venous blood, but this is not practical for large numbers in this study.[37] We observed fever in participants with microfilaremia less frequently than expected from the smaller safety trials. [8–10] This may have been due to reduced intensity of monitoring in our study and the absence of other febrile infectious syndromes including malaria. We noted that women were more likely to report an AE despite being less likely to have microfilaremia. This gender difference has been reported in other studies [10, 13], and is hypothesised to be because women are more likely to report subjective AEs than men.

There are a number of strengths to our study. First, the randomisation of villages achieved balance of treatment groups with regard to baseline demographic characteristics, prevalence of filariasis and scabies, numbers of people treated and monitored for AEs. Second, we maintained high participation over 7 days of safety monitoring, with only 14 participants lost to follow-up. It is unlikely that these participants experienced a severe or serious AE that went undetected due to the close proximity of the study team to the local health care providers.

Our study was limited by the inability to blind participants and assessors to the treatment group. In a safety study of this kind, blinding would have been ideal, but was not feasible,

**Table 3.  Risk factor analysis for occurrence of adverse events.**

| Factor | | Total[a] | | | Univariate analysis[a] | | Multivariate analysis[a,b] | |
|---|---|---|---|---|---|---|---|---|
| | | N | n | % | RD % | (95% CI) | RD % | (95% CI) |
| Treatment | DA | 1213 | 203 | 16.7 | Ref | | Ref | |
| | IDA | 2342 | 393 | 16.8 | 0.1 | (-5.2 to 5.4) | 0.4 | (-4.8 to 5.5) |
| Island | Rotuma | 1525 | 254 | 16.7 | Ref | | Ref | |
| | Gau | 2030 | 342 | 16.8 | 0.2 | (-4.9 to 5.3) | 1.5 | (-3.4 to 6.4) |
| Gender | Male | 1853 | 286 | 15.4 | Ref | | Ref | |
| | Female | 1702 | 310 | 18.2 | 2.8 | (-0.1 to 5.7) | 3.5 | (0.9 to 6.1) |
| Age (years) | 2–4 | 156 | 11 | 7.1 | Ref | | Ref | |
| | 5–9 | 537 | 58 | 10.8 | 3.7 | (-0.5 to 7.8) | 3.2 | (-1.5 to 7.9) |
| | 10–14 | 526 | 61 | 11.6 | 4.5 | (-0.1 to 9.1) | 4.5 | (-0.3 to 9.3) |
| | 15–24 | 453 | 77 | 17.0 | 9.9 | (4.1 to 15.7) | 9.7 | (3.6 to 15.9) |
| | 25–34 | 393 | 80 | 20.4 | 13.2 | (8 to 18.5) | 11.9 | (6.5 to 17.3) |
| | 35–49 | 671 | 143 | 21.3 | 14.2 | (9.7 to 18.7) | 11.7 | (6.5 to 17) |
| | 50–64 | 557 | 119 | 21.4 | 14.3 | (7.9 to 20.7) | 12.5 | (5.4 to 19.6) |
| | ≥65 | 262 | 47 | 17.9 | 10.8 | (5.1 to 16.6) | 9.1 | 2.7 to 15.4) |
| LF | Total CFA negative | 3065 | 462 | 15.1 | Ref | | - | - |
| | Total CFA positive | 490 | 134 | 27.3 | 13.4 | (7.8 to 19.1) | - | - |
| | CFA1—weak positive | 165 | 35 | 21.2 | 7 | (-0.4 to 14.3) | - | - |
| | CFA2—medium positive | 147 | 40 | 27.2 | 13.4 | (5.1 to 21.8) | - | - |
| | CFA3—strong positive | 178 | 59 | 33.1 | 19.4 | (10.5 to 28.4) | - | - |
| LF | Microfilariae positive | 139 | 60 | 43.2 | 28.2 | (20.6 to 35.7) | 26.4 | (18.5 to 34.3) |
| Scabies | Identified | 461 | 67 | 14.5 | -2.6 | (-7 to 1.8) | 1.3 | (-2.8 to 5.5) |
| STH[c] | Positive microscopy | 165 | 23 | 13.9 | -4.9 | (-10.6 to 0.8) | - | - |

RD: risk difference; CI: confidence interval; Ref: reference group; DA: diethylcarbamazine and albendazole; IDA: ivermectin, diethylcarbamazine and albendazole; LF: lymphatic filariasis; CFA: circulating filarial antigen; STH: soil-transmitted helminths.

[a] Denominator for total, univariate and multivariate analyses' N = 3555 (excluded 21 declined CFA testing, 11 unreadable Mf smears, 11 declined Mf smears).

[b] Multivariate risk difference analysis excludes CFA scores and STH results.

[c] Denominator for STH analysis N = 881.

largely because of the addition of a topical treatment to the intervention, as well as the logistics of providing a second dose to all participants. A further limitation of the study is the relatively small number of participants in our sub-analyses, meaning that cautious interpretation is required for risk differences.

Current modelling suggests that MDA coverage above 65% is required for success of the strategy to interrupt transmission.[38, 39] After more than 10 rounds of MDA for filariasis in the study sites, the risk of participation fatigue increases and therefore likelihood of high MDA coverage decreases.[40, 41] Comprehensive community engagement is recognised as crucial for success of the introduction of the new IDA strategy.[14] In our experience, highlighting the action of the IDA MDA on the three common infections found at our study sites prior to enrolment, assisted in our high population coverage for MDA. This message can now be coupled with the findings of the study that the combination of the three drugs for MDA is safe.

Our study describes the large burden of neglected tropical diseases in Fiji that may benefit from IDA. Our results confirm that introducing MDA using IDA as an elimination strategy for lymphatic filariasis in Fiji is as safe as the current DA combination that has been implemented and accepted by communities since 2002. Our results provide confidence for the safety

of IDA within other Pacific populations where lymphatic filariasis remains endemic, along with scabies and soil-transmitted helminths.

## Supporting information

**S1 Protocol. Community Based Safety of 2-drug (Diethylcarbamazine and Albendazole) versus 3-drug (Ivermectin, Diethylcarbamazine and Albendazole) Therapy for Lymphatic Filariasis in Fiji–Protocol v6.0 6[th] August 2019.**
(PDF)

**S1 Fig. Randomisation, treatment and monitoring flowchart.** DA: diethylcarbamazine and albendazole; IDA1: ivermectin one dose, diethylcarbamazine and albendazole; IDA2: ivermectin two dose with DA; AE: adverse event.
(PDF)

**S2 Fig. Adverse event monitoring flowchart.** AE: adverse event; PI: principal investigator; HREC: human research ethics committee. [a] Passive monitoring: participants asked to seek out study monitors if grade 1 symptoms worsen or new symptoms develop. All participants with symptoms higher than grade 1 will be actively followed.
(PDF)

**S1 Table. Medication dosing schedule.**
(PDF)

**S2 Table. Adverse event grading chart.**
(PDF)

**S3 Table. CONSORT 2010 checklist of information to include when reporting a cluster randomised trial.**
(PDF)

**S4 Table. Population and participant demographics by village.** DA: diethylcarbamazine and albendazole; IDA1: ivermectin one dose, diethylcarbamazine and albendazole; IDA2: ivermectin two dose with DA; IQR: interquartile range.
(PDF)

**S5 Table. Demographics of non-participants in 2017.** DA: diethylcarbamazine and albendazole; IDA1: ivermectin one dose, diethylcarbamazine and albendazole; IDA2: ivermectin two dose with DA; Pop.: population; Med.: median; IQR: interquartile range.
(PDF)

**S6 Table. Baseline parasitic infection prevalence and adverse events by village.** DA: diethylcarbamazine and albendazole; IDA1: ivermectin one dose, diethylcarbamazine and albendazole; IDA2: ivermectin two dose with DA; IQR: interquartile range; LF: lymphatic filariasis; CFA +: circulating filarial antigen positive; Mf +: microfilariae positive; STH: soil- transmitted helminths; AE: adverse event. [a] Denominator tested for CFA excludes n = 60 declined and n = 93 ineligible. [b] Denominator for Mf excludes n = 11 unreadable smears, n = 71 declined and n = 93 ineligible. [c] Denominator for AE excludes n = 200 not treated and n = 14 lost to follow-up.
(PDF)

**S7 Table. List of reported adverse event symptoms with associated severity in order of frequency in participants treated for filariasis and followed up.** AE: adverse event; DA: diethylcarbamazine and albendazole; IDA: ivermectin, diethylcarbamazine and albendazole; Mod:

moderate severity; NEC: not elsewhere classified.
(PDF)

## Acknowledgments

We thank all participants from Rotuma and Gau islands, Fiji. We acknowledge the support of the Fiji Ministry of Health and Medical Services (FMOHMS), the Fiji Ministry of iTaukei Affairs, the Fiji Ministry of Education, Heritage & Arts, and the Rotuman Council. In addition to the named authors the following people made significant contributions to the study: Ravi Naidu, physician, Colonial War Memorial Hospital, FMOHMS; Mikaele Lutumailagi, Medical Officer, Eastern Division, FMOHMS; Luke Ravula, Medical Officer, Eastern Division, FMOHMS; Humphrey Biutilomaloma, Fiji Data Manager, Murdoch Children's Research Institute (MCRI); Aminiasi Koroivueti, Fiji Project Officer, MCRI; Patrick Lammie, The Taskforce for Global Health, Andrew Majewski, The Taskforce for Global Health; Joshua Bogus, Global Project Manager, Death to Onchocerciasis and Lymphatic Filariasis (DOLF), St. Louis; Katiuscia O'Brian, Global Data Manager, DOLF, St. Louis; Catherine Bjerum, Laboratory and Good Clinical Practice Trainer, Case Western Reserve University; and the rest of the Fiji Integrated Therapy team.

## Author Contributions

**Conceptualization:** Josaia Samuela, Gary J. Weil, Andrew C. Steer.

**Data curation:** Myra Hardy.

**Formal analysis:** Myra Hardy, Anneke C. Grobler, Andrew C. Steer.

**Funding acquisition:** Gary J. Weil, Andrew C. Steer.

**Investigation:** Myra Hardy.

**Methodology:** Myra Hardy, Christopher L. King, Gary J. Weil, John M. Kaldor, Andrew C. Steer.

**Project administration:** Myra Hardy, Josaia Samuela, Mike Kama, Meciusela Tuicakau, Lucia Romani, Andrew C. Steer.

**Resources:** Mike Kama, Christopher L. King, Gary J. Weil.

**Software:** Myra Hardy.

**Supervision:** Josaia Samuela, Margot J. Whitfeld, Leanne J. Robinson, John M. Kaldor, Andrew C. Steer.

**Validation:** Anneke C. Grobler.

**Visualization:** Myra Hardy.

**Writing – original draft:** Myra Hardy.

**Writing – review & editing:** Myra Hardy, Josaia Samuela, Mike Kama, Meciusela Tuicakau, Lucia Romani, Margot J. Whitfeld, Christopher L. King, Gary J. Weil, Anneke C. Grobler, Leanne J. Robinson, John M. Kaldor, Andrew C. Steer.

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
