## [Decision Letter · Decision Letter 0]

27 Nov 2019

Dear Dr. Hardy:

Thank you very much for submitting your manuscript "The safety of combined ivermectin, diethylcarbamazine and albendazole in the setting of multiple neglected tropical diseases: a cluster randomised trial in Fiji" (#PNTD-D-19-01467) for review by PLOS Neglected Tropical Diseases. Your manuscript was fully evaluated at the editorial level and by independent peer reviewers. The reviewers appreciated the attention to an important problem, but raised some substantial concerns about the manuscript as it currently stands. These issues must be addressed before we would be willing to consider a revised version of your study. We cannot, of course, promise publication at that time.

We therefore ask you to modify the manuscript according to the review recommendations before we can consider your manuscript for acceptance. Your revisions should address the specific points made by each reviewer. 

When you are ready to resubmit, please be prepared to upload the following:

(1) A letter containing a detailed list of your responses to the review comments and a description of the changes you have made in the manuscript.

(2) Two versions of the manuscript: one with either highlights or tracked changes denoting where the text has been changed (uploaded as a "Revised Article with Changes Highlighted" file); the other a clean version (uploaded as the article file).

(3) If available, a striking still image (a new image if one is available or an existing one from within your manuscript). If your manuscript is accepted for publication, this image may be featured on our website. Images should ideally be high resolution, eye-catching, single panel images; where one is available, please use 'add file' at the time of resubmission and select 'striking image' as the file type. 

Please provide a short caption, including credits, uploaded as a separate "Other" file. If your image is from someone other than yourself, please ensure that the artist has read and agreed to the terms and conditions of the Creative Commons Attribution License at http://journals.plos.org/plosntds/s/content-license (NOTE: we cannot publish copyrighted images). 

(4) If applicable, we encourage you to add a list of accession numbers/ID numbers for genes and proteins mentioned in the text (these should be listed as a paragraph at the end of the manuscript). You can supply accession numbers for any database, so long as the database is publicly accessible and stable. Examples include LocusLink and SwissProt.

(5) To enhance the reproducibility of your results, we recommend that you deposit your laboratory protocols in protocols.io, where a protocol can be assigned its own identifier (DOI) such that it can be cited independently in the future. For instructions see http://journals.plos.org/plosntds/s/submission-guidelines#loc-methods

While revising your submission, please upload your figure files to the Preflight Analysis and Conversion Engine (PACE) digital diagnostic tool, https://pacev2.apexcovantage.com/ PACE helps ensure that figures meet PLOS requirements. To use PACE, you must first register as a user. Then, login and navigate to the UPLOAD tab, where you will find detailed instructions on how to use the tool. If you encounter any issues or have any questions when using PACE, please email us at figures@plos.org.

We hope to receive your revised manuscript by Jan 26 2020 11:59PM. If you anticipate any delay in its return, we ask that you let us know the expected resubmission date by replying to this email.

To submit a revision, go to https://www.editorialmanager.com/pntd/ and log in as an Author. You will see a menu item call Submission Needing Revision. You will find your submission record there. 

Sincerely,

Sabine Specht

Associate Editor

Jennifer Keiser

Deputy Editor

Reviewer's Responses to Questions

**Key Review Criteria Required for Acceptance?**

**Methods**

-Are the objectives of the study clearly articulated with a clear testable hypothesis stated?

-Is the study design appropriate to address the stated objectives?

-Is the population clearly described and appropriate for the hypothesis being tested?

-Is the sample size sufficient to ensure adequate power to address the hypothesis being tested?

-Were correct statistical analysis used to support conclusions?

-Are there concerns about ethical or regulatory requirements being met?

Reviewer #1: The objectives are clearly articulated and the population appropriate for these objectives. The objectives of the paper are to evaluate the safety of IDA in Fiji, providing a detailed account of AEs following MDA in Fiji and an analysis of factors associated with their occurrence including filariasis, scabies, and STH infections. However, this subset of data from Fiji as part of the international study fundamentally compromises the power of the analysis to achieve its objectives of establishing if IDA materially alters the safety profile of DA. While there are unique co-infection rates in Fiji, the approach of the authors to answer safety would have been more informative if the analysis was based on coinfections rather than national boundaries and included data from the entire study. 

There are several other aspects of the methods employed for the main study and this sub-group analysis that are limiting for the stated objectives: 

1. there are two fundamental issues missing from the rationale:

A. There is no discussion of the pharmacological aspects of adding ivermectin to this regimen to understand impact on metabolic pathways, risk of drug:drug interactions or transporter interactions, and extrinsic factors such as food effects. A statement on the pharmacology of the three agents would be the minimum expected

B. An evaluation of the impact on these drugs on pathogen/host respoonse to treatments to contextualise the safety data or approach taken is also not presented to justify the methods chosen. 

These evaluations would inform the safety methods employed, including laboratory assessments if required, and the nature and duration of adverse events to be of particular interest

2. 2 days of active and 5 days of passive adverse event follow up should have been justified in the paper, particularly passive reporting as a method to fully elucidate safety differences. It appears practicality and cost driven rather than the optimal approach to establishing relative risk. As a result, the adverse event rates appear unexpectedly low for systematically collected data, even over such a short period of time. 

3. the rationale for not using a standardised toxicity grading scale to ensure consistency of grading across multiple sites, and the training provided for safety assessors to ensure standardisation of approach is also not described 

4. It should be stated if the analysis plan was signed off prior to database lock and unaltered post-lock, and that statistical consideration was made for this sub-group analysis. 

5. The rationale for not blinding the study has also not been made and is a significant compromise to this design. 

6. Contraceptive measures for women of child bearing potential should be described. 

7. It is assumed that a standard question was asked to elicit adverse events from participants. This should be stated. 

8. The training provided to study nurses to ensure consistency between reviewers and grading of events should also be described. Did they all attend formalised training at an investigator meeting? Was this applied internationally? 

9. Were concurrent medications recorded?

10. Was regulatory approval received? 

11. How were the variable requirements for fed and fasting state handled in the protocol? 

The statistical testing performed should comply with ICHE9. A statement about compliance with Declaration of Helsinki would also be expected.

Reviewer #2: Specific Points:

1) Line 145: this would be a good section to describe the age inclusion/exclusion differences from the parent study.

2) Lines 363-4: need the numbers of pruritus in <5 year old group give the comment about increased pruritis in the IDA group if disproportionately in the children <5 would be worth noting given they may have scabies not optimally treated with topicals and without ivermectin.

Reviewer #3: The authors should clarify and delineate between the safety and efficacy on the main objective of the study. More insight and clarity is needed in understanding the study methods, the specific procedures involved in both intervention and data collection, how responses are transcribed and translated, how issues of language barriers had been addressed.

Authors have not pointed out specific hypothesis being tested.

**Results**

-Does the analysis presented match the analysis plan?

-Are the results clearly and completely presented?

-Are the figures (Tables, Images) of sufficient quality for clarity?

Reviewer #1: The rationale for the microfilariae levels chosen to report against should be provided. Mf presence or absence is the analysis of interest for adverse events incidence and severity but the number of infected people is small which limits the conclusions that can be drawn from the study. 

There are too many subdivisions and subgroups in the demography. Suggest this is simplified and standardised to 0-28 day, 28 day to 2 year, 3 to 11 years, 12 to 17, 18 to 65, >65 for age and few subdivisions for Mf unless justified by the biology of response. Height and weight should have been collected and should be reported. 

In line 318, it appears that 100 people who were not eligible for ivermectin received it. Under what circumstances and what happened to them? 

Caution should be expressed from the relative risk analysis in Table 3. There are too few patients and the risk of Type 1 or 2 errors is not accounted for in the authors presentation. 

Line 216 described drug/pathogen interactions as important for adverse events but it is more accurate to describe this as pathogen/host interactions as a result of drug efficacy. 

Line 357 p value, if post hoc, should be declared as post hoc. 

Outcomes of SAEs are not given.

Reviewer #2: The ITT reporting of the age 2-5 group as having received ivermectin needs to be supplemented with an analysis of the "as treated" population given the pre-specification to treat the only >5 y/o children with ivermectin. This may or may not have an impact on analyzing the pruritus finding in the two regimens. 

Table 2: suggest either an Asterix on the IDA column for the age 2-4 year old group or remove the numbers from the column given they did not receive IDA as per protocol exclusion.

Reviewer #3: The tables provides substantial information on the findings of the study. However the tables could be summarized further to present the most essential information, the demographic information can be captured in the text.

Footnote on key descriptions of the table should be provided. The discussion should elaborate on key issues of safety identified . The reported AEs should be exhaustively discussed to allay doubt etc.

**Conclusions**

-Are the conclusions supported by the data presented?

-Are the limitations of analysis clearly described?

-Do the authors discuss how these data can be helpful to advance our understanding of the topic under study?

-Is public health relevance addressed?

Reviewer #1: The conclusions are not supported. There is Insufficient data in this sub-analysis and fundamental questions about the main study that undermine the objectives.

Reviewer #2: The conclusion of the parent study that triple drug combination is comparable in safety to the two drug standard of care is supported by the size of the study and the results presented in the multinational publication cited (Weil et al PLoS 2019.) In Fiji, adverse events were frequent albeit largely mild with only 3 unrelated SAE’s. The results reported from Fiji do not demonstrate an effect of scabies or STH co-endemic infections on adverse events and this is important information for the endemic population of Fiji that remains resistant to MDA of the 2 drug regimen.

Reviewer #3: More information and discussion is required on how the data can be of hep to advance our understanding safety in this study. The limitations of the analysis needs to come out clearly. 

What level of confidence can we have in the finding that the results or findings in DA group compared to IDA, is not due to some form bias. How wide the gap is the confidence intervals as to make a strong case for judging the similarity between the two groups.

**Editorial and Data Presentation Modifications?**

Reviewer #1: The paper is well written.

Reviewer #2: (No Response)

Reviewer #3: Minor revision

**Summary and General Comments**

Reviewer #1: It is too ambitious to turn this national analysis into a declaration of safe or not safe for the new regimen. The study was not double blinded, and yet it could have been, the follow up duration was inadequate and should have driven by the biology of drug/host and pathogen/host factors. Further, non-standard methods of collecting adverse events and grading them were employed.

Reviewer #2: The authors report on the safety of the standard of care regimen of diethylcarbamazine (DEC) and albendazole (DA regimen) versus the standard of care plus two doses of ivermectin (IDA1 and IDA2 regimen) and baseline status of scabies or soil-transmitted helmint infections (STH) from Fiji that apparently was part of a multi-national trial of the two regimens for treatment of lymphatic filariasis (LF). Fiji is endemic for LF and notably has previously received 10 rounds pf MDA with DA. 

Clarification is needed between the design of the multi-national study that included Fiji and this manuscript from Fiji alone, now submitted and under review. The funding source and grant are the same yet the age of inclusion of children is different ( >5 in Weil et al, >2 in Hardy et al) and the total numbers differ (3,419 in Weil et al and 3612 in Hardy et al) that can not be reconciled by the 179 children enrolled in Hardy et al between 2-5 years of age. 

The reporting of the data could also be more informative with further explanation of the as treated population. While an ITT analysis is appropriate to report safety results to avoid any interpretations based on post-randomisation events, the fact that children <5 were not treated per protocol design with ivermectin indicates that the as treated population should also be reported given ivermectin assignment is determined a priori by age. 

My suggestion is to present the analysis as originally described in the SAP but to add the “as treated data” for the <5 particularly in the sub-group with scabies since the next older age group had the largest burden of scabies and the numbers could be meaningful. If in the end the original conclusions are un-changed, this would strengthen the conclusions of safety of the three-drug regimen in Fiji.

Reviewer #3: The study is novel in the identification of the key issue of safety of combined therapy. The authors though needs to clearly exhaust this in the discussion to satisfy reader quest for information Give that cases of AEs have been reported, those cases all and all are critical to the outcome of the study. They must be exhaustively discussed and all issues clarified. What are the AEs caused by and what can we rely on to strengthen our belief in this.

PLOS authors have the option to publish the peer review history of their article (what does this mean?). If published, this will include your full peer review and any attached files.

Reviewer #1: No

Reviewer #2: No

Reviewer #3: Yes: Yaya Camara

---

## [Decision Letter · Decision Letter 1]

31 Jan 2020

Dear Dr. Hardy,

We are pleased to inform you that your manuscript 'The safety of combined triple drug therapy with ivermectin, diethylcarbamazine and albendazole in the neglected tropical diseases co-endemic setting of Fiji: a cluster randomised trial' has been provisionally accepted for publication in PLOS Neglected Tropical Diseases.

Before your manuscript can be formally accepted you will need to complete some formatting changes, which you will receive in a follow up email. A member of our team will be in touch within two working days with a set of requests.

Best regards,

Sabine Specht

Associate Editor

Jennifer Keiser

Deputy Editor

Reviewer's Responses to Questions

**Key Review Criteria Required for Acceptance?**

**Methods**

-Are the objectives of the study clearly articulated with a clear testable hypothesis stated?

-Is the study design appropriate to address the stated objectives?

-Is the population clearly described and appropriate for the hypothesis being tested?

-Is the sample size sufficient to ensure adequate power to address the hypothesis being tested?

-Were correct statistical analysis used to support conclusions?

-Are there concerns about ethical or regulatory requirements being met?

Reviewer #1: This is a retrospective analysis of a substudy and, as such, is compromised. In terms of design, the authors have done their best to walk the tight rope between justifying their approach and acknowledging its inherent limitations.

Reviewer #2: (No Response)

Reviewer #3: The methods of the study are in conformity with the results and the authors made efforts to address concerns from the first review.

**Results**

-Does the analysis presented match the analysis plan?

-Are the results clearly and completely presented?

-Are the figures (Tables, Images) of sufficient quality for clarity?

Reviewer #1: It would be helpful to have the investigator's assessment of causality for the serious adverse events stated. It is often helpful to present safety data not just by incidence, but by severity (i.e toxicity grading scale) and investigator assessment of causality as well. It gives the reviewer a better perspective of the importance of what is seen.

Reviewer #2: (No Response)

Reviewer #3: The results sections is in conformity with the CONSORT (Consolidated Reporting Of trials) and much clearer for a general grasp of the findings.

**Conclusions**

-Are the conclusions supported by the data presented?

-Are the limitations of analysis clearly described?

-Do the authors discuss how these data can be helpful to advance our understanding of the topic under study?

-Is public health relevance addressed?

Reviewer #1: The new language of the conclusions strikes a very reasonable compromise.

Reviewer #2: (No Response)

Reviewer #3: A reasonable conclusion have been made as have been supported by the data. The authors can improve on the discussion by giving examples of similar studies and contrast their finding with this whether in agreement or not. Good Luck

**Editorial and Data Presentation Modifications?**

Reviewer #1: None.

Reviewer #2: (No Response)

Reviewer #3: Minor Revision

**Summary and General Comments**

Reviewer #1: the paper provides supportive rather than definitive data on the question of whether or not co-morbidities affect the safety profile. This is useful and helpful information for the field and this should be published. This study was not prospectively designed to address the question being asked and thus was not powered appropriately or blinded. The authors have done a good job of softening the conclusions to be more appropriate to the protocol but a retrospective analysis on a subset with post hoc analyses is not how we should be answering these important questions.

Reviewer #2: Addressed major comments.

Reviewer #3: This version of the article is much better written and its objectives are very reflected in both the Methodology and the Results. The methods and results sections are in conformity with the CONSORT (Consolidated Reporting Of trials) statement's extension to cluster randomized trials guidelines thereby making the study transparent and reproducible. The approach to analysis is also appropriate.

Best wishes,

PLOS authors have the option to publish the peer review history of their article (what does this mean?). If published, this will include your full peer review and any attached files.

Reviewer #1: No

Reviewer #2: No

Reviewer #3: Yes: Yaya Camara

---

## [Editor Report · Acceptance letter]

10 Mar 2020

Dear Dr. Hardy,

We are delighted to inform you that your manuscript, "The safety of combined triple drug therapy with ivermectin, diethylcarbamazine and albendazole in the neglected tropical diseases co-endemic setting of Fiji: a cluster randomised trial," has been formally accepted for publication in PLOS Neglected Tropical Diseases.

Best regards,

Serap Aksoy

Editor-in-Chief

Shaden Kamhawi

Editor-in-Chief
